# Preparation and Characterization of Nifedipine-Loaded Dry Medium Internal-Phase Emulsions (Dry MIPEs) to Improve Cellular Permeability

**DOI:** 10.3390/pharmaceutics14091849

**Published:** 2022-09-01

**Authors:** Sukannika Tubtimsri, Yotsanan Weerapol

**Affiliations:** Faculty of Pharmaceutical Sciences, Burapha University, Chonburi 20131, Thailand

**Keywords:** nifedipine, dry emulsion, medium internal-phase emulsions

## Abstract

A nifedipine (NP) dry emulsion was fabricated by the adsorption of medium internal-phase emulsions (MIPEs). Simple homogenizers were first used to mix conventional liquid MIPEs, and then a microfluidizer was used to reduce the resulting emulsions’ droplet sizes. The dry MIPEs (solid) were produced by adsorbing the emulsions onto solid carriers with a high surface area. The dry MIPEs were diluted in a simulated gastric fluid under gentle agitation to form emulsions. The diluted dry MIPEs were divided into three groups based on an NP content of 0.3%, 0.5%, and 0.7%, with sizes of 5026–5404 nm, 2583–3233 nm, and 1318–1618 nm in diameter, respectively. Powder X-ray diffraction (PXRD) measurements and differential scanning calorimetry (DSC) were used to characterize the physical properties of the dry MIPEs. The samples contained 0.5% or 0.7% drug, 2–4% surfactant, and 8–16% oil (5RH2/8, 7RH2/8, and 7RH4/16) and showed the characteristic peak for NP. No NP peak was observed in formulations with 0.3% NP and any oil-phase content (3RH2/8, 3RH4/16, and 3RH8/32). The formulations with 0.5% drug, 4–8% surfactant, 16–32% oil (5RH4/16 and 5RH8/32) and those with 0.7% drug, 8% surfactant, and 32% oil (7RH8/32) also did not show the peak for NP. These findings demonstrated that microfluidization improved the solubility of NP in the formulations. The subsequent drug dissolution results were consistent with the DSC thermogram and PXRD pattern results. 3RH2/8, 3RH4/16, 3RH8/32, 5RH4/16, 5RH8/32, and 7RH8/32 were completely dissolved and showed higher dissolved NP amounts than 5RH2/8, 7RH2/8, 7RH4/16, and NP powder. The lowest mean dissolution time was for 7RH8/32 (13.31 ± 0.87 min). Caco-2 cells were used to determine drug uptake, and 7RH8/32 showed the maximum intracellular uptake (10.89%). After storage under accelerated and normal conditions (3 and 6 months), the selected formulations remained stable. The developed formulations can be used to improve NP solubility and absorption.

## 1. Introduction

New drug candidates often have poor water solubility, which leads to poor oral absorption, high fluctuating bioavailability, and nonproportionality when adjusting doses. Increasing the water solubility of a drug can be performed in several ways, including salt formation, drug particle reduction, and complexation. An effective and easy way to increase drug water solubility is to develop a formulation that helps increase solubility and improves drug absorption [1,2]. Preparations in the form of emulsions are another way to reduce the dissolution time of lipophilic drugs and increase oral absorption. Lipophilic drugs are soluble in the oil phase (discontinuous phase) of emulsion systems and are absorbed into the body through the gastrointestinal tract more rapidly with greater bioavailability than the unmodified drug [3]. A previous study used scallop-gonad isolated proteins as a food emulsifier to create emulsion-based delivery systems to enhance the bioavailability of the test drug [4]. The effects of carrier oils on ***β***-carotene cellular absorption also have been studied. In intestinal epithelial cells, cellular absorption of ***β***-carotene was greater for an emulsion than for the unmodified form. In another study, a Caco-2 cell culture model was used to study the effect of the oil phase on vitamin-E bioavailability. Oil-in-water emulsions enriched with ***α***-tocopherol acetate were created by combining a natural emulsifier (*Quillaja* saponin) with triglycerides as lipids [5]. Vitamin-E absorption was higher in the lipid emulsions because of the higher solubilization in the triglyceride-based system.

An emulsion is generally a liquid state, which presents problems with storage and administration of the drug to the patient. Liquid emulsion preparation has certain drawbacks, such as transportation, storage, and particularly, an instability that can occur in several ways, such as phase inversion, Ostwald ripening, creaming, sedimentation, and coalescence [6]. It is difficult to produce emulsions in a solid dosage form in the pharmaceutical industry, so they must be prepared in soft gelatin form [7]. Liquid emulsions can be converted to solid dry emulsions by adsorbing them onto solid carriers [1,8]. After administration, the dry emulsions spontaneously disperse in gastric fluid and form a simple emulsion. The limitation of dry emulsion preparation is that the amount of solubilized drug in the emulsion is low. Regarding solidification, absorption using lipid-base systems (without water) instead of emulsions is possible, but the most common problem is incomplete drug release, which may be due to the high viscosity of the lipid-base systems making the distribution of oil droplets difficult and inconsistent [9,10]. Excessive exposure of the solids carriers to the liquid lipid-base systems will result in the formation of an organogel.

Nifedipine (NP) was chosen as a model poorly water-soluble drug in this experiment. NP is a calcium-channel blocker used in the treatment of cardiovascular disorders. According to its solubility and bioavailability properties, NP is in the Biopharmaceutics Categorization System class II [11]. Because of its low water solubility (5 μg/mL), its dissolution and subsequent oral absorption are limited [12].

Therefore, our researchers considered reducing the viscosity of the lipid-base formulation and increasing its dispersion on the adsorbent carrier by preparing an emulsion with a medium internal phase. Medium internal-phase emulsions (MIPEs) were considered (oil-phase volume ratio of 10–40%) to increase the amount of drug in a low soluble, limited fat-soluble formulation [13]. The study’s aim was to develop dry MIPEs through a new process. A microfluidizer was used to prepare nanosized emulsion droplets for adsorption onto a nonporous and high surface area solid carrier (amorphous fumed silica [FS]). The dry emulsions spontaneously dispersed and formed a simple emulsion under gentle agitation. The drug dissolution of test emulsions was compared by evaluating the various ratios of drug, oil, surfactant, and water. Cellular uptake evaluation was performed to assess NP absorption performance.

## 2. Materials and Methods

Caprylic/capric glyceride (oil) was obtained from Sasol, Germany. Aerosil^®^ 200 (FS) was bought from Evonik. Polyoxyl 40 hydrogenated castor oil (surfactant) was supported by BASF, Thailand. The test drug NP was acquired from Xilin Pharmaceutical Raw Material Co., Ltd., Xilin, China. NP was powdered and passed through a sieves-size 80# before being used. As NP is light sensitive, all samples were kept wrapped in aluminum foil or amber-colored containers during the whole experimental process. Methanol and acetonitrile used in the analytical method were acquired from Acros Organics, Belgium. Caco-2 was obtained from the American Type Culture Collection and produced from colorectal adenocarcinoma epithelial cells (catalog no. HTB-37). Analytical grade types of all chemical substances were used in this research.

### 2.1. Preparation of Emulsion

Table 1 shows emulsion formulations consisting of drug, surfactant, and oil. The name of the formulations refers to the drug content as the first number (3, 5, and 7), which is the percentage of NP (0.3%, 0.5%, and 0.7%) and the percentages of surfactant and oil as the second and third numbers, respectively. In our preliminary study, we used a surfactant to oil ratio of 1:4. In addition, various dry emulsions with 0.3% NP (3RH2/8, 3RH4/16, and 3RH8/32) without microfluidization were prepared, and the coalescence of emulsion was visually observed during the drying process. Microfluidization was performed to create smaller emulsions and provide greater emulsion stability [14]. In brief, NP was added to a mixture of a surfactant and oil followed by addition of water before homogenization. To prepare the coarse emulsions, a homogenizer (Polytron, Kinematika AG Littau, Switzerland) was used with a mixing rate of 15,000 rpm for 10 min in a water bath (10 °C). The coarse emulsions were collected and homogenized 50 times at 100 MPa in a microfluidizer (NV-200-D, Nanomizer, Japan). The particle/droplet sizes of the coarse emulsions were measured by laser light scattering (PD-10S, Nikkiso, Tokyo, Japan). The nanosizes of emulsions were measured by photon-correlation spectroscopy (model Zetasizer Nano ZS, Malvern, UK).

A high surface area and a nonporous adsorbent, FS, were chosen to eliminate porosity factors that could lead to reduced drug release [15]. FS was mixed with the nanoemulsion at a ratio of 1:1 (weight of FS: weight of lipid phase of the emulsion) and then dried in an oven at 40 °C for 48 h. The obtained dry MIPEs were stored in a desiccant chamber. In these formulations, the dry MIPEs were diluted, and the emulsion spontaneously formed in the aqueous solution. To measure the size of the dry MIPEs after dispersion, the sample was diluted (199-fold) under gentle agitation with simulated gastric fluid (SGF) without pepsin and left for 2 h. Prior to measuring the emulsion droplet sizes, the FS in the dispersed emulsion was removed by centrifugation (700× *g*) for 10 min. Emulsion droplets were investigated under an optical microscope (CX41, Olympus, Tokyo, Japan) and a polarized filter (CX-AL, Olympus).

### 2.2. Morphological Examination

A scanning electron microscope (model LEO1450VP, Carl Zeiss Microscopy GmbH, Munich, Germany) with a 10-keV acceleration voltage was adjusted to examine the morphology of the samples. The sample powders were adhered onto a stub using double-sided sticky tape. Before inspection, all the samples were vacuumed and then gold-coated.

### 2.3. Analysis of NP

High-performance liquid chromatography (HPLC; model Jasco PU-2089 with a model Jasco UV-2070 plus multi-wavelength UV–VIS detector, Jasco, Tokyo, Japan) was used to analyze NP. The separation and analysis procedures were as described in a previous report [16]. A C18 (5 µm, 4.6 × 250 mm) ACE^®^ column was used to separate and quantify the NP concentration. The detection wavelength was 235 nm, and the isocratic flow rate of the mobile phase was 1 mL/min. Before use, a mobile phase solution of 50% methanol: acetonitrile (1:1) in water was filtered through a 0.45-µm nylon membrane filter and then degassed in a sonicating bath. A 20-μL injection volume was used for all samples in the HPLC. ChromNav software (Jasco, Japan) was used to calculate the NP peak area. The HPLC analysis was performed with triplicate samples.

### 2.4. Powder X-ray Diffractometry (PXRD)

PXRD measurements were performed with an angle speed of 4°/min from 5–45° (MiniFlex II, Rigaku, Tokyo, Japan), a voltage of 30 kV, and a current of 15 mA using Cu Kα radiation.

### 2.5. Differential Scanning Calorimetry (DSC)

Thermograms of the dry MIPEs, NP, and physical mixtures (PMs) were analyzed by DSC (model DSC 8000, Perkin Elmer, Waltham, MA, USA). A precise sample weight of 2.5 mg was crimped into an aluminum DSC liquid pan type. Subsequently, the sample pan was heated at a rate of 10 °C/min.

### 2.6. Porosimetry Examination

A surface area and pore size analyzer were used to determine the surface area of the dry MIPEs (Model Nova 2000e, Quantachrome, Boynton Beach, FL, USA). To eliminate remaining water, samples were degassed at 100 °C for 2 h under vacuum. At −196 °C (77 K) adsorption and desorption isotherms were recorded. The Berret–Joyner–Halenda (BJH) technique was used to compute the surface area.

### 2.7. Dissolution Study

The dissolution of NP was accomplished using dissolution apparatus I (PharmaTest, Germany). Samples (equivalent to NP 10 mg) were placed in each of three baskets, and the dissolution medium was SGF without pepsin (900 mL, pH 1.2), and they were protected from light. A rotating paddle at a speed of 50 rpm was used for stirring. Dissolution samples (5 mL) were collected at 5, 10, 15, 30, 60, 90, and 120 min from the dissolution vessels and passed through 0.45-µm nylon membrane filters. A fresh medium (5 mL) was added to maintain a dissolution-sink condition. HPLC was used to measure the amount of NP dissolved in the medium.

The mean dissolution time (*MDT*) was calculated using the dissolution data to determine the extent of dissolved NP improvement from the different dry MIPEs according to the following equation [17]:$$MDT\;=\;\frac{\sum_{i=0}^{n}\bar{t_{i}}\Delta Q_{i}}{\sum_{i=0}^{n}\Delta Q_{i}}$$
where $\bar{t_{i}}$ is the midpoint of the time period during which the fraction Δ*Q_i_* of the drug has been dissolved from the formulation, *i* is the number of dissolution samples, and *n* is the number of time points for dissolution sampling.

### 2.8. Evaluating Cellular Uptake of NP

The performance of cellular uptake from the emulsions was also evaluated by using Caco-2 cells, which were seeded at a density of 2 × 10^4^ cells per well in a 24-well plate and incubated for 24 h at 37 °C with 5% CO_2_. Caco-2 cells were treated with 200 g/mL of NP powder in the 3RH2/8, 5RH4/16, and 7RH8/32 emulsions (below cytotoxicity concentration). The treated media was removed after 3 h [18], and the cells were washed three times with 1 mL of phosphate-buffered saline, then lysed with 1 mL lysis buffer composed of 10 mM Tris-HCl, 150 mM NaCl, 1% TritonX-100, 1 mM EDTA, and 0.1% sodium dodecyl sulfate. After 0.5-h, 0.1 mL of cell lysate was collected into a microcentrifuge tube to which 0.9 mL of methanol was added. The microcentrifuge tubes were centrifuged, and the supernatant from the microcentrifuge tubes was collected for further measurement of drug content by HPLC under the same conditions as in the analysis of NP.

### 2.9. Stability of NP

To evaluate the stability of NP, all dry MIPEs (*n* = 3) were stored for 3 or 6 months under accelerated (40 °C/75% relative humidity) and ambient (25 °C) conditions. The NP concentration, emulsion droplet size after diluting in SGF, and drug dissolution after storage (6 months) using the difference factor ($f_{1})$ and similarity factor $(f_{2}$) [19] to assess the dry MIPEs’ stability. The relative error between the two profiles is measured by the ($f_{1})$ factor, which computes the percentage difference between the two dissolution profiles at each time point:$$f_{1}=\left(\frac{\sum_{t=1}^{n}\left|R_{t}-T_{t}\right|}{\sum_{t=1}^{n}R_{t}}\right)\times 100$$

A measurement of the degree of similarity in the percentage dissolution between the two profiles, the $(f_{2}$) factor is a logarithmic reciprocal square root transformation of the sum of the squared error:$$f_{2}=50\times \log_{10}\left[\frac{100}{\sqrt{1+\frac{\sum_{t=1}^{n}{\left(R_{t}-T_{t}\right)}^{2}}{n}}}\right]$$
where n denotes the number of time points, $R_{t}$ denotes the average rate of dissolution for the initial day products at time *t*, and $T_{t}$ denotes the average rate of dissolution for the test product at that time. When the test and initial day product profiles are the same, the ($f_{1})$ value is equal to zero. When the test and initial day product profiles are identical, the $(f_{2}$) value is equal to 100.

### 2.10. Statistical Analysis

SPSS version 28.0 for Windows (IBM Corp., Armonk, NY, USA) was used to perform analysis of variance and Levene’s test for variance homogeneity. If the result of Levene’s test was insignificant or significant, *post hoc* testing (*p* < 0.05) for multiple comparisons was performed using Scheffé or Games–Howell test, respectively.

## 3. Results and Discussion

The coarse emulsion droplets ranged in size from 3000 to 6700 nm, and the polydispersity index (PI) ranged from 0.455 to 0.680 (Table 2). The emulsion droplet sizes were dramatically reduced by microfluidization. Nanosized emulsion droplets were obtained (122.7–255.7 nm), which may be because of the high energy of the microfluidization and the sufficient amounts of surfactant used to cover all of the oil droplets thoroughly [20,21]. However, the presence of a high PI of the emulsion may have been caused by an unstable thermodynamic system [22].

In screening test, the emulsions were passed through the microfluidizer 20, 30, 40, 50, 60, and 70 passes, which gave small emulsion sizes that did not change after over 50 passes (Appendix A). In addition, the input pressure was kept low because of the viscosity of the mixture and because it has also been previously reported that applying high pressure would result in high lipid oxidation [20].

The surface area test results of dry MIPEs are shown in Table 3. The surface area of FS was 213 m^2^/g, and when FS was absorbed by the emulsion, the surface area of the dry MIPEs decreased. In addition, the surface area decreased with increasing amounts of liquid absorbed (increasing the amount of surfactant and oil from 2% to 8% and 8% to 32%, respectively). This is consistent with previous studies [23] that show that when a higher amount of liquid was absorbed on the adsorbent, the surface area was lower. Regardless of the NP dose of 0.3%, 0.5%, or 0.7%, when using the same amount of surfactant and oil, the surface area did not significantly change (*p* < 0.05).

Scanning electron microscopy (SEM) images of dry MIPEs, FS, and NP are shown in Figure 1. NP had a smooth surface and a rectangular shape, whereas the FS particles had a rough surface, as shown by the SEM images. Surface roughness particles and scattered angular-shaped particles, similar to those seen in NP, were observed in the 5RH2/8, 7RH2/8, and 7RH4/16 emulsion formulations and could have formed by NP crystallizing and detaching from the carriers. SEM images of the 3RH2/8, 3RH4/16, 3RH8/32, 5RH4/16, 5RH8/32, and 7RH8/32 formulations showing particles with surface roughness. The resulting images showed NP evenly distributed throughout the dry MIPEs carrier. In a previous study, a 30% amount of carrier was shown to be sufficient for producing an excellent free-flowing powder [24,25]. In the current study, we used an adsorbent level of 50%, and the dry emulsion that resulted was suitable.

### 3.1. Thermal Analysis

Figure 2 shows the DSC thermograms of the NP, dry MIPEs, and PMs. The samples referred to as 3PMs, 5PMs, and 7PMs were PMs consisting of 8% surfactant and 32% oil, with NP percentages of 0.3%, 0.5%, and 0.7%, respectively. The thermograms of NP showed sharp endothermic peaks at 173 °C [26]. The 3PM, 5PM, and 7PM exhibited low-intensity endothermic peaks coinciding with NP, indicating that NP had not changed. Similar results were found in the PMs contained 0.3%, 0.5% or 0.7% drug, 2–4% surfactant, and 8–16% of oil (data not shown). The 3RH2/8, 3RH4/16, 3RH8/32, 5RH4/16, 5RH8/32, and 7RH8/32 formulations showed no endothermic peaks. The thermal behavior of NP changed, as evidenced by the DSC thermograms. This finding indicates that NP could be a dispersed form at the molecular level in dry MIPEs adsorbed onto a carrier [27]. The endothermic peaks of low intensity in the 5RH2/8, 7RH2/8, and 7RH4/16 formulations matched the NP peak, showing that insoluble NP was present in the dry MIPEs.

### 3.2. PXRD Analysis

The diffraction patterns of the NP, dry MIPEs, and PMs are shown in Figure 3. The NP diffraction peaks showed sharp high-intensity peaks at 2θ of 10°, 11°, 19°, and 24°, indicating that the intrinsic NP was crystalline [26]. The PMs had minor NP crystalline and FS peaks, indicating no changes in the NP crystallinity diffraction. Similar findings were observed in the PMs contained 0.3%, 0.5% or 0.7% drug, 2–4% surfactant, and 8–16% of oil (data not shown). The low peaks of the PXRD pattern at 5RH2/8, 7RH2/8, and 7RH4/16, corresponding to NP, showed that NP was present in the dry MIPEs, and this finding was consistent with the findings from the DSC thermograms and SEM images. There was no peak for NP in the PXRD patterns and DSC thermograms of the 3RH2/8, 3RH4/16, 3RH8/32, 5RH4/16, 5RH8/32, and 7RH8/32, suggesting molecular dispersion of NP in the dry MIPEs.

The theoretical values of NP solubilization in the formulations consisting of various surfactant/oil percentages (2/8, 4/16, and 8/32) were calculated, and the percentages of the solubilized NP should be 0.16%, 0.32%, and 0.92% for the corresponding samples [22,25]. The maximum NP solubilization percentages of 3RH2/8 and 5RH4/16 were higher than the theoretical values. These results were possibly due to the nanosized emulsion formation achieved by the microfluidization, as previously reported [25,28]. According to that research, higher experimental solubility values than the theoretical values clearly demonstrated that increasing the nanoparticle interface area enables higher solubility.

### 3.3. Droplet-Size Analysis after Dilution

The rate and extent of in vivo drug absorption are affected by droplet diameter, which is a significant factor in emulsification performance. A smaller emulsion droplet size provides for faster drug dissolution and a greater interface area, which increases drug absorption [29,30]. In these intended formulations, the dry MIPEs should be easily dispersed after dilution. The average sizes and images from optical microscope of dry MIPEs after dilution under gentle agitation in SGF are shown in Table 3 and Appendix A. Emulsion droplet sizes were divided into three groups based on the NP content of 0.3%, 0.5%, and 0.7%, with sizes of 5026–5404 nm, 2583–3233 nm, and 1318–1618 nm, respectively. All dry MIPEs had slightly high PI (0.511–0.622). After centrifugation to precipitate the FS, the emulsion droplet size remained unaltered. The results of this study showed that the NP content in the emulsion affected the emulsion size after dilution, possibly due to the increased dose of the drug affecting the hydrophobicity and surface tension in the oil droplets. Drug content has previously been shown to have an effect on emulsion size [31,32]. Regarding particle size, when a hydrophilic drug (lidocaine) was added to the formulations, the droplet size increased relative to the size in the blank formulations. These results were due to the increase in the interfacial tension caused by the interaction of surfactants and hydrophilic drugs.

A study of emulsions composed of lipophilic drugs showed that the drugs were solubilized in the oil phase and accumulated in the interfacial region [33]. In the current study, increasing NP may have increased the accumulation of drug at the interface, leading to curvature disruption and resulting in smaller emulsion droplets. These aspects suggest interferences at the interface that may lead to a change in the interfacial tension. The results suggest that the emulsion diameter is affected by the amount of drug and its properties in the formulation [32].

### 3.4. In Vitro Dissolution of NP

The dissolution of NP is shown in Figure 4. The developed dry MIPEs were intended for an immediate-release formulation and therefore almost 100% of the NP was completely dissolved after 120 min. This result showed that dry MIPEs led to higher drug release than NP powder. The 3RH2/8, 3RH4/16, 3RH8/32, 5RH4/16, 5RH8/32, and 7RH8/32 emulsions showed that NP was completely dissolved and had higher percentages of dissolved NP than the 5RH2/8, 7RH2/8, and 7RH4/16 emulsions and plain NP powder. The NP in 7RH2/8, 7RH4/16, and 5RH2/8 was partially dissolved at 120 min, and the NP dissolved from the dry MIPEs increased in ascending order of 7RH2/8, 7RH4/16, and 5RH2/8. The results of the in vitro NP dissolution are supported by the X-ray diffractogram and DSC thermogram data. The NP in the 3RH2/8, 3RH4/16, 3RH8/32, 7RH8/32, 5RH4/16, and 5RH8/32 emulsions was completely dissolved in the oil phase. However, the NP in the 7RH2/8, 7RH4/16, and 5RH2/8 emulsions was partially dissolved and provided higher NP dissolution than the NP powder. Some parts of the NP were unchanged, which was a very insoluble form (in water) and insoluble in the oil phase. Incomplete dissolution profiles were caused by unchanged NP. This study demonstrated the need to consider the amount of oil phase in which the NP is suitably soluble. Incomplete release of the drug due to the use of solid carriers has previously been reported. The spontaneous emulsion powder for solubility enhancement of a poorly water-soluble drug was fabricated by physical adsorption of the test drug onto a solid carrier. Experimental results showing incomplete dissolution have previously been reported as caused by the agglomeration of particles and poor dispersion in the dissolution medium [34]. In the present study, particle agglomeration may have been a secondary factor related to low drug dissolution. The main factor was that the drug was incompletely soluble in the oil phase and crystallization occurred [35,36]. In Figure 5, the samples with complete NP solubility were used to determine the MDT. Among the formulations of 7RH8/32, 5RH4/16, 5RH8/32, 3RH2/8, 3RH4/16, and 3RH8/32, the calculated MDT values in descending order were 13.31 ± 0.87 min, 16.10 ± 0.67 min, 20.42 ± 0.24 min, 23.49 ± 2.82 min, 26.14 ± 3.01 min, and 29.55 ± 1.11 min, respectively. There were no significant differences in MDT between 7RH8/32, 5RH4/16, and 5RH8/32. The MDT was significantly lower in 7RH8/32 than in 3RH2/8, 3RH4/16, and 3RH8/32. The MDT of 5RH4/16 was not significantly different from those of 5RH8/32 and 3RH2/8 but was significantly lower than those in 3RH4/16 and 3RH8/32. Interestingly, the MDTs for 5RH4/16 were not significantly different among the group with complete dissolution. MDT determines the dissolution of the drug from the dosage form as well as its controlled release ability. The lowest dissolution rate of drug from the dosage form is indicated by a greater MDT [17]. As a result, the formulation has a slower onset of action and stronger drug-retention ability. 7RH8/32 had the shortest MDT, implying that it may dissolve rapidly and have a rapid onset of action. 7RH8/32, 5RH4/16, and 5RH8/32 were selected for further experimentation.

### 3.5. Evaluating Cellular Uptake of NP

The permeability of drugs is critical for achieving pharmacological effectiveness. In the case of oral administration, the proportion of drug absorbed across the intestinal wall determines whether the drug concentration in the blood circulation is sufficient to have a therapeutic impact on the target organ [37]. Cellular uptake by Caco-2 cells was chosen in this study to investigate the permeability of the drug from preparations. Figure 6 shows the percent cellular absorption of 5RH4/16, 5RH8/32, and 7RH8/32 NP powders after 3 h of exposure to each formulation. For 7RH8/32, 5RH4/16, 5RH8/32, and NP powder, the percentages of cellular uptake were 10.89 ± 0.85%, 5.61 ± 0.77%, 4.86 ± 0.57%, and 0.34 ± 0.01%, respectively. The maximum percentage of cellular uptake was observed in the 7RH8/32 treatment group and was 32 times greater than that of the NP powder. These findings were in accordance with their in vitro dissolution results and diluted dry MIPE size results. The 7RH8/32 emulsion had the maximum drug solubility and the smallest droplet size and was rapidly absorbed via passive transport through Caco-2, resulting in a high bioavailability [38].

### 3.6. Stability of Dry MIPEs

During the study periods of 3 and 6 months, the dry MIPEs revealed no visible physical alterations. Before the stability test, the NP content of 7RH8/32, 5RH4/16, 5RH8/32, 3RH2/8, 3RH4/16, and 3RH8/32 was approximately 100% (Table 4). Under both conditions, the NP content was >99% at the end of 3 and 6 months. In the formulations that were examined, there were no substantial NP losses. DSC and PXRD were used to study the physicochemical properties of the chosen dry MIPEs after storage. DSC indicated no endothermic peaks that corresponded to the intrinsic peaks of NP in the chosen formulations after storage under accelerated conditions (Appendix A). PXRD patterns of the selected formulations showed halo-like patterns under both storage conditions, indicating that there was no crystalline NP peak in those formulations (Appendix A). Similar results (DSC and PXRD analysis) were found under ambient conditions. The stability of the selected dry MIPEs was assessed by diluting them 199-fold in SGF and then determining the emulsion size. The emulsion sizes of 3RH2/8, 3RH4/16, and 3RH8/32 were still below 5500 nm; 5RH4/16, 5RH8/32 were still below 3500 nm; and 7RH8/32 was still below 1700 nm. The PI ranges obtained were comparable to those obtained on the first day. After 6 months of storage in both settings, the dissolution properties of the selected formulations were satisfactory and identical to those of the initial day preparations (Appendix A). The dissolution profiles were compared using the difference factor and similarity factor. After 6 months of storage under both conditions, the difference and similarity factor values of 5RH4/16, 5RH8/32, and 7RH8/32 were near 0 and 100, respectively, compared to that on the initial day. These findings revealed how well the formulations preserved the initial dissolution characteristics following storage under both conditions.

## 4. Conclusions

We successfully developed dry MIPEs, which were prepared by microfluidization and adsorbed onto amorphous FS in the prepared formulations. The solubilized NP in the formulations depended on the concentrations of the surfactants and oils. The very low solubility of NP affected the drug dissolution characteristics of the formulation. The dry MIPEs spontaneously emulsified by dilution with the dissolution medium, and the identified suitable formulations were 5RH4/16, 5RH8/32, and 7RH8/32. The concentration of oil and surfactant in each formulation should provide adequate drug solubility in the oil phase of the emulsion. The solubility of the NP in the formulation was improved by microfluidization. The size of the emulsion droplets was affected by the amount of drug and its properties in the formulation. Uptake of NP by Caco-2 cells was also evaluated, and the highest absorption was 10.89% by 7RH8/32. After 3 and 6 months of storage under accelerated and normal conditions, the selected formulations were stable. The developed formulations can be used to improve NP solubility and absorption.

## Figures and Tables

**Figure 1 pharmaceutics-14-01849-f001:**
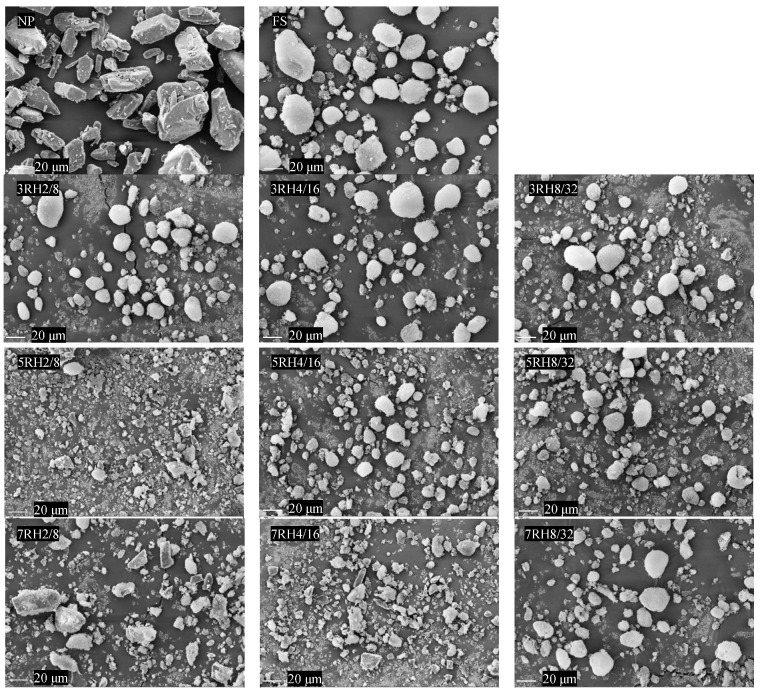
Scanning electron microscope (SEM) pictures of FS, NP, and dry medium internal-phase emulsions (dry MIPEs).

**Figure 2 pharmaceutics-14-01849-f002:**
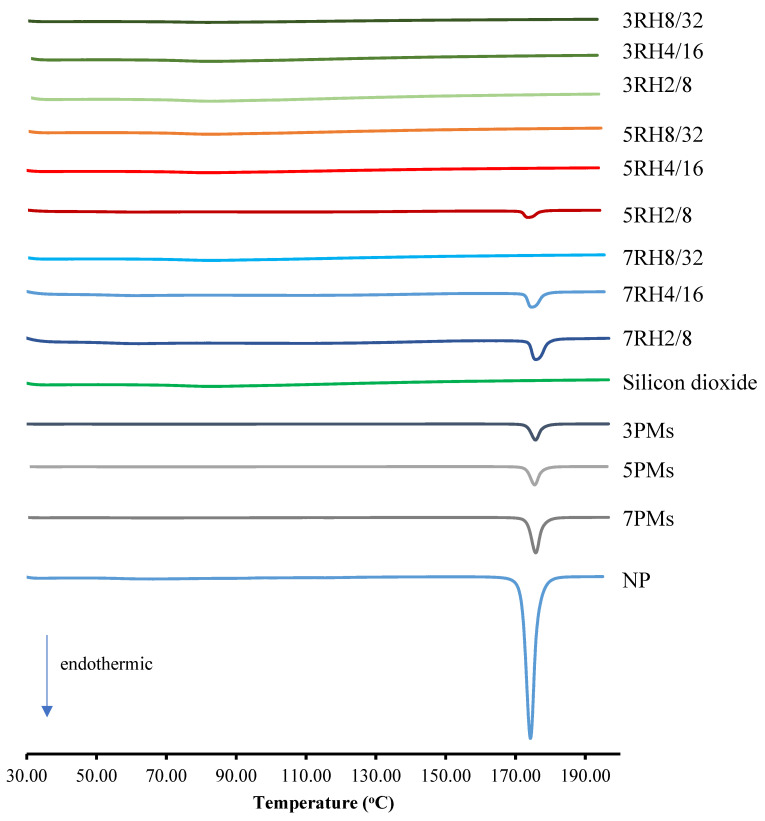
Differential scanning calorimetry (DSC) thermograms of the dry MIPEs and NP powder. The 3PMs, 5PMs, and 7PMs were physical mixtures (PMs) of 8% surfactant and 32% oil, with NP contents of 0.3%, 0.5%, and 0.7%, respectively.

**Figure 3 pharmaceutics-14-01849-f003:**
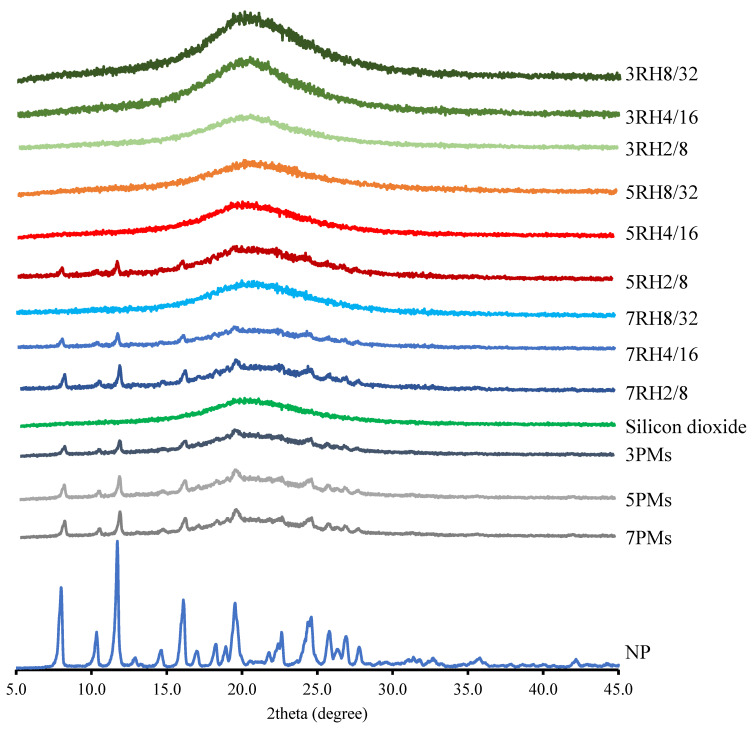
Powder X-ray diffraction patterns of the dry MIPEs and NP powder. The 3PMs, 5PMs, and 7PMs were PMs of 8% surfactant and 32% oil, with NP contents of 0.3%, 0.5%, and 0.7%, respectively.

**Figure 4 pharmaceutics-14-01849-f004:**
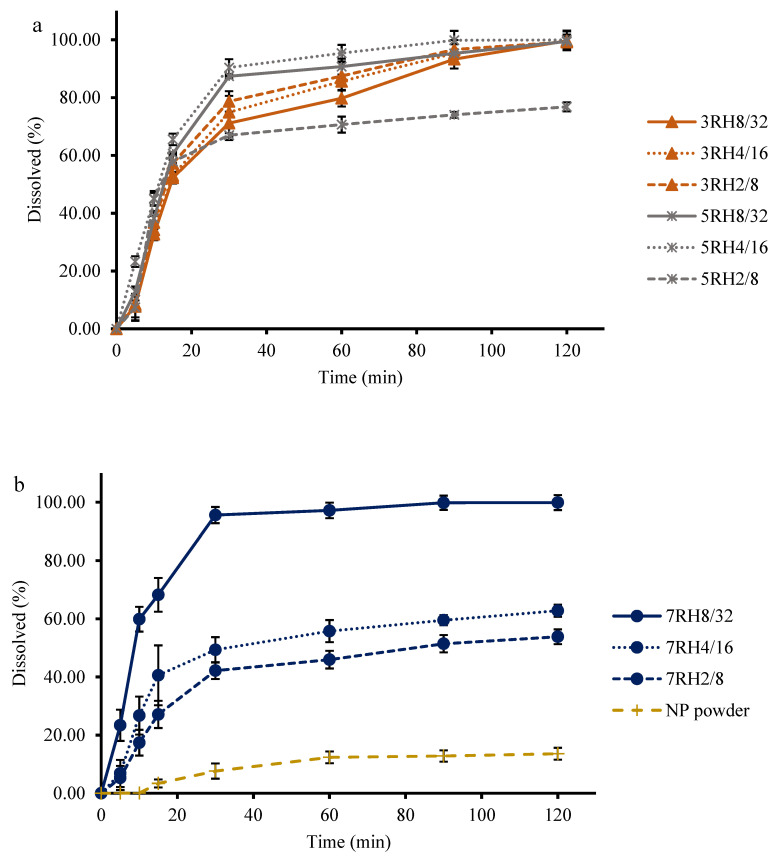
Dissolution of dry NP powder in (**a**) 3RH2/8, 3RH4/16, 3RH8/32, 5RH2/8, 5RH4/16, and 5RH8/32 and in (**b**) 7RH2/8, 7RH4/16, 7RH8/32, and NP powder alone.

**Figure 5 pharmaceutics-14-01849-f005:**
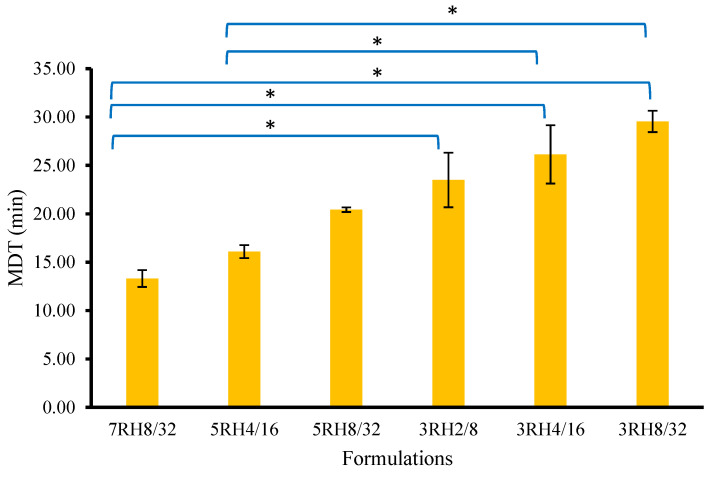
The mean dissolution times (MDTs) of the selected dry MIPEs (3RH2/8, 3RH4/16, 3RH8/32, 5RH4/16, 5RH8/32, and 7RH8/32) and NP powder (* *p* < 0.05).

**Figure 6 pharmaceutics-14-01849-f006:**
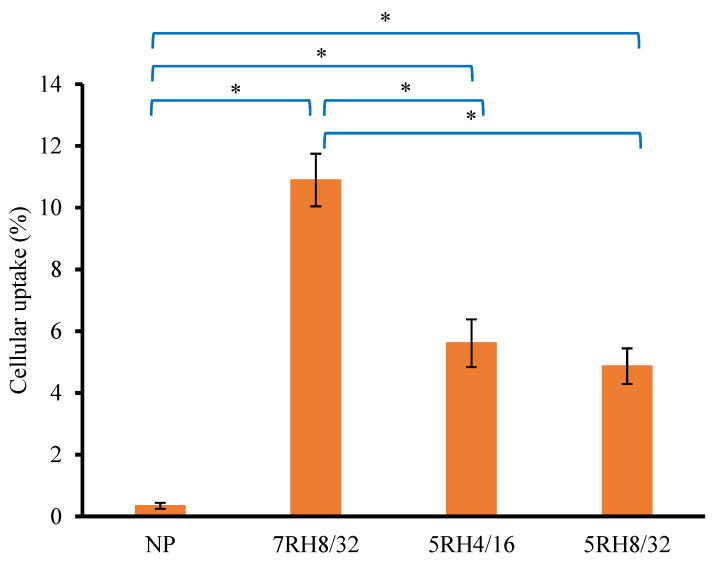
Percent cellular absorption of 5RH4/16, 5RH8/32, and 7RH8/32 NP powders after 3 h of exposure to each formulation. (*n* = 3) (* *p* < 0.05).

**Table 1 pharmaceutics-14-01849-t001:** Amounts of drug, surfactant, and oil in the emulsion formulations.

Formulations	NP (%)	Surfactant (%)	Oil (%)
3RH2/8	0.3	2	8
3RH4/16	0.3	4	16
3RH8/32	0.3	8	32
5RH2/8	0.5	2	8
5RH4/16	0.5	4	16
5RH8/32	0.5	8	32
7RH2/8	0.7	2	8
7RH4/16	0.7	4	16
7RH8/32	0.7	8	32

**Table 2 pharmaceutics-14-01849-t002:** Emulsion droplet sizes after passing though the simple homogenizer and microfluidizer (*n* = 3).

Formulations	Simple Homogenizer (nm [PI])	Microfluidizer (nm [PI])
3RH2/8	6721.0 ± 345.1 [0.680]	182.3 ± 42.1 [0.342]
3RH4/16	6319.9 ± 210.2 [0.531]	164.3 ± 32.2 [0.323]
3RH8/32	6313.2 ± 207.5 [0.525]	157.7 ± 30.3 [0.310]
5RH2/8	6540.7 ± 120.1 [0.504]	150.2 ± 47.1 [0.401]
5RH4/16	4427.0 ± 352.3 [0.549]	121.2 ± 13.1 [0.463]
5RH8/32	3080.1 ± 542.1 [0.483]	118.7 ± 10.2 [0.405]
7RH2/8	6403.3 ± 242.8 [0.490]	255.7 ± 39.8 [0.311]
7RH4/16	3652.2 ± 254.2 [0.455]	204.7 ± 29.0 [0.357]
7RH8/32	3221.7 ± 542.9 [0.557]	122.7 ± 25.2 [0.352]

values are expressed as mean ± standard deviation. PI: poly dispersion index.

**Table 3 pharmaceutics-14-01849-t003:** Surface areas and droplet diameters of the dry medium internal-phase emulsions (dry MIPEs) after dilution.

Formulations	BJH Surface Area (m^2^/g)	Size of Emulsion after Dilution (nm [PI])
3RH2/8	169.2 ± 8.1 *	5404.4 ± 244.3 [0.524]
3RH4/16	145.4 ± 5.5 *	5297.7 ± 214.4 [0.622]
3RH8/32	108.2 ± 7.9 *	5026.4 ± 213.3 [0.545]
5RH2/8	163.1 ± 5.8 *	3233.2 ± 338.3 [0.521]
5RH4/16	143.3 ± 6.4 *	2804.4 ± 313.3 [0.512]
5RH8/32	100.5 ± 8.1 *	2583.0 ± 323.2 [0.524]
7RH2/8	160.1 ± 6.2 *	1618.6 ± 424.1 [0.514]
7RH4/16	140.7 ± 5.9 *	1562.8 ± 452.1 [0.554]
7RH8/32	99.7 ± 8.5 *	1318.4 ± 424.0 [0.511]

values are expressed as mean ± standard deviation. PI: poly dispersion index. the surface area of amorphous fumed silica (FS) was 213 ± 7.6 m^2^/g. * *p* < 0.05 compared with FS.

**Table 4 pharmaceutics-14-01849-t004:** Remaining percentages of NP after storage under accelerated conditions (relative humidity 40 °C/75%) and ambient (25 °C) (*n* = 3).

Formulations	Day 0 (%)	3 Months (%)		6 Months (%)	
		Accelerated Condition	Ambient Condition	Accelerated Condition	Ambient Condition
NP powder	100.05 ± 0.15	100.03 ± 0.13	100.02 ± 0.11	100.02 ± 0.09	100.03 ± 0.10
3RH2/8	100.12 ± 0.25	100.01 ± 3.27	100.02 ± 2.37	100.22 ± 0.17	100.40 ± 3.11
3RH4/16	100.21 ± 0.13	100.03 ± 2.12	100.01 ± 1.43	100.01 ± 1.17	100.12 ± 0.69
3RH8/32	100.31 ± 0.23	100.01 ± 2.26	100.01 ± 1.37	100.03 ± 1.53	100.03 ± 0.88
5RH4/16	100.11 ± 0.33	100.12 ± 2.22	100.08 ± 1.02	100.00 ± 2.01	100.12 ± 0.79
5RH8/32	100.07 ± 1.20	100.20 ± 1.72	100.02 ± 1.05	100.01 ± 1.03	100.13 ± 0.78
7RH8/32	100.13 ± 2.26	100.11 ± 1.53	100.02 ± 1.93	101.03 ± 1.31	100.10 ± 0.71

values are expressed as the mean ± standard deviation.

## Data Availability

All data are included in the manuscript.

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
