# Peer review of "Preparation and Characterization of Nifedipine-Loaded Dry Medium Internal-Phase Emulsions (Dry MIPEs) to Improve Cellular Permeability"

_pharmaceutics, 2022, doi:10.3390/pharmaceutics14091849_

Round 1

Reviewer 1 Report (Previous Reviewer 1)

In the present manuscript entitled “Preparation and Characterization of Nifedipine-loaded Dry Medium Internal-phase Emulsions (Dry MIPEs) to Improve Cellular Permeability” authors have reported the dry emulsion to ultimately improve the oral bioavailability which has been mimicked by performing the cellular permeability in the present submission. This is the second time I am reviewing this manuscript and I don’t see much change in my previous comments. I am not sure if the authors have chosen the right challenge to work on as nifedipine shows more than 90% bioavailability following the oral administration and is commercially available in the immediate release as well as extended release dosage form in tablet form for once in a day administration. I am not sure by designing the so-called Dry MIPEs which current problem can be overcome. Besides the poor rationale to justify the work there are many flaws/contradictions in the results which are as follows;

1.     The sizes presented in Table 2 are not matching with the sizes presented in the corresponding SEM images. There seems to be a huge difference in the size between these two. Also, the figure's indicators are not clearly visible.

2.     For most of the parameters in stability studies data is not shown which is very important to critically evaluate the author's findings.

3.     For PXRD analysis authors must have used the equivalent quantity of NP to have the comparison.

4.     What is the basis of performing the cellular uptake for 3 h when the dissolution studies were performed for 2 h? If the drug is getting dissolved in just half an hour (as per the author's findings) then for exact relation it could be better to perform these studies for 2 hours.

5.     Why the cellular uptake was performed only for 4 groups while all the groups were included in other studies? If authors finalized the 7RH8/32 as the optimized formulation then only this could be compared with NP.

6.     Figure 5 shows the mean dissolution time for different groups including NP powder however MDT for NP powder is missing in the said figure.

7.     In table 4 authors have included different formulations for stability testing. Including NP alone could be a good idea to exactly have the contribution of the developed formulations on the NP stability.

8.     As per the statement, in vivo drug absorption is affected by droplet diameter. The authors have not performed any in vivo study in the present submission. If we consider the cellular uptake as a possible substitute then also it is not conclusive as the used formulations for this study have almost the same particle size.

Line 115 the dry MIPEs were administered orally and the emulsions spontaneously formed. In the continuation, the authors have mentioned that dry MIPEs were diluted with SGF to measure the size after dispersion. Sentences are creating confusion as I am not sure if this study was performed in-vivo or in-vitro. I am not able to find the data following the dispersion in SGF mentioned in lines 116-119.

Author Response

In the present manuscript entitled “Preparation and Characterization of Nifedipine-loaded Dry Medium Internal-phase Emulsions (Dry MIPEs) to Improve Cellular Permeability” authors have reported the dry emulsion to ultimately improve the oral bioavailability which has been mimicked by performing the cellular permeability in the present submission. This is the second time I am reviewing this manuscript and I don’t see much change in my previous comments. I am not sure if the authors have chosen the right challenge to work on as nifedipine shows more than 90% bioavailability following the oral administration and is commercially available in the immediate release as well as extended release dosage form in tablet form for once in a day administration. I am not sure by designing the so-called Dry MIPEs which current problem can be overcome. Besides the poor rationale to justify the work there are many flaws/contradictions in the results which are as follows;

Response: We thank the reviewer for their valuable comments. Nifedipine was used as a model of a poorly water-soluble drug in this experiment. According to its solubility and bioavailability, NP is classified by the Biopharmaceutics Categorization System as class II. Because of its low water solubility, its dissolution and subsequent oral absorption are limited (lines 6670). Indeed, the commercial formulation of immediate-release nifedipine is a solution in a soft gelatin capsule; our study intended to develop an immediate-release powder formulation. We needed to replace the liquid dosage form with a solid version and improve the solubility compared to the unmodified powder of the model drug.

  1. The sizes presented in Table 2 are not matching with the sizes presented in the corresponding SEM images. There seems to be a huge difference in the size between these two. Also, the figure's indicators are not clearly visible.

Response: In Table 2, we have presented the emulsion size in the preparation process of dry MIPEs. The SEM images (Figure 1) show the dry MIPEs as finished products.

  1. For most of the parameters in stability studies data is not shown which is very important to critically evaluate the author's findings.

Response: More details have been added in the supplemental files in this submission (Figure S3, S4, and S5). Line 388, 390 and 397.

  1. For PXRD analysis authors must have used the equivalent quantity of NP to have the comparison.

Response: Thank you for your valuable advice. The quantity of NP was equal to each formulation in the analysis of PXRD and DSC.

  1. What is the basis of performing the cellular uptake for 3 h when the dissolution studies were performed for 2 h? If the drug is getting dissolved in just half an hour (as per the author's findings) then for exact relation it could be better to perform these studies for 2 hours.

Response: The in vitro dissolution of dry MIPEs was performed in 2 h because we needed to confirm the dispersion ability of the emulsification in the stomach. The cellular uptake assessing the performance of the developed formulation in the intestine took 3 h. Additionally, from a literature review, we found that the incubation period should be 3 h for the cellular uptake of an emulsion [1, 2].

  1. Why the cellular uptake was performed only for 4 groups while all the groups were included in other studies? If authors finalized the 7RH8/32 as the optimized formulation then only this could be compared with NP.

Response: The samples selected for the cellular uptake assay have been described in the section “In vitro dissolution of NP,” which is briefly described as follows. The incomplete dissolution formulations were excluded. The MDT was computed, and the lowest MDT was 7RH8/32, while 5RH4/16 and 5RH8/32 did not have a significant difference compared to that of the MDT of 7RH8/32. Finally, 7RH8/32, 5RH4/16, and 5RH8/32 were chosen for the cellular uptake study.

  1. Figure 5 shows the mean dissolution time for different groups including NP powder however MDT for NP powder is missing in the said figure.

Response: For present the performance of the suitable formulations. In Figure 5, the formulation with complete NP solubility were used to determine the MDT. Therefore, the incomplete NP solubility of NP powder was excluded for this calculation and the Figure.

  1. In table 4 authors have included different formulations for stability testing. Including NP alone could be a good idea to exactly have the contribution of the developed formulations on the NP stability.

Response: Thank you for your valuable advice. The results have been added in this revision. (Table 4)

  1. As per the statement, in vivo drug absorption is affected by droplet diameter. The authors have not performed any in vivo study in the present submission. If we consider the cellular uptake as a possible substitute then also it is not conclusive as the used formulations for this study have almost the same particle size.

Response: The droplet size was analyzed after the dilution to confirm the dispersibility of dry MIPEs and the size of the emulsion. To confirm the size of the emulsion for this response, dry MIPEs were diluted with 200 mg/mL of NP powder in media (Table R1) for the cellular uptake study. The sizes of the emulsions were similar to those in Table 3. These results may confirm that the size of the emulsion has not changed due to the dilution.

In fact, we agree with the reviewer that an in vivo experiment would be useful. However, we were unable to conduct this experiment in our laboratory at this time. Please accept our apology.

Table R1 shows the size of the selected dry MIPEs after dilution (200 mg/mL)

Formulations

Size of emulsion after dilution (nm [PI])

5RH4/16

2783.1 ± 307.5 [0.501]

5RH8/32

2545.4 ± 342.1 [0.494]

7RH8/32

1359.3 ± 410.3 [0.495]

values are expressed as mean ± standard deviation

PI: poly dispersion index

Line 115 the dry MIPEs were administered orally and the emulsions spontaneously formed. In the continuation, the authors have mentioned that dry MIPEs were diluted with SGF to measure the size after dispersion. Sentences are creating confusion as I am not sure if this study was performed in-vivo or in-vitro. I am not able to find the data following the dispersion in SGF mentioned in lines 116-119.

Response: We apologize for the mistake, and the manuscript has been modified in revised version on line 115-116.

[1]      Sermkaew, N.; Wiwattanawongsa, K.; Ketjinda, W.; Wiwattanapatapee, R. Development, Characterization and Permeability Assessment Based on Caco-2 Monolayers of Self-Microemulsifying Floating Tablets of Tetrahydrocurcumin. AAPS PharmSciTech, 2013, 14 (1), 321–331. https://doi.org/10.1208/s12249-012-9912-2.

[2]      Zaichik, S.; Steinbring, C.; Friedl, J. D.; Bernkop-Schnürch, A. Development and In Vitro Characterization of Transferrin-Decorated Nanoemulsion Utilizing Hydrophobic Ion Pairing for Targeted Cellular Uptake. Journal of Pharmaceutical Innovation, 2021. https://doi.org/10.1007/s12247-021-09549-2.

Reviewer 2 Report (Previous Reviewer 3)

In my opinion, questions have not been answered appropriately.

Claims concerning the pharmaceutical nature of the formulation are not solved. If the aim of the authors is the formulation of a free-flowing powder it never can be an emulsion. No experimental demonstrations are given that an emulsion is initially achieved and that an emulsion is in-situ spontaneously generated after dilution.

Sizes of globules are needed instead of particle sizes. A rapid method is an optical microscopy observation. In this stage silica is not present and this possible source of  interference would be obviated.

Optical microscopy with polarized light is  the most easy-and-rapid to perform technique to  demonstrate the existence of crystals, although in the presence of amorphous solids . Other  techniques are obviously more complex and it is non-sense to apply them without this preliminar testing.

Answer about the uncomplete dissolution do not solve the uncertainty: this fact is dued to a poor solubility or that a formulation  makes a cluster-effect retaining part of the drugdose.

Author Response

In my opinion, questions have not been answered appropriately.

Response: We thank the reviewer for their valuable comments.

Claims concerning the pharmaceutical nature of the formulation are not solved. If the aim of the authors is the formulation of a free-flowing powder it never can be an emulsion. No experimental demonstrations are given that an emulsion is initially achieved and that an emulsion is in-situ spontaneously generated after dilution.

Response: The method to achieve the spontaneous emulsion is the size measurement of the dry MIPEs after dispersion that explained in line 116-119. The results of emulsion size after dilution were presented in Table 3, the discussions wereexplained in line 299-317.

From your valuable advice, emulsion droplets of dry MIPEs (finished products) after dilution with simulated gastric fluid (SGF) were investigated under a polarized light optical microscope (line 120-121) and the images from optical microscopy were added in supplement material to confirm the spontaneous emulsification (Figure S2). Line 298-299.

Sizes of globules are needed instead of particle sizes. A rapid method is an optical microscopy observation. In this stage silica is not present and this possible source of  interference would be obviated.

Response: After the microfluidizer process of emulsion, we have presented the emulsion size (Table 2) in the preparation process of dry MIPEs. The insoluble solid carrier (silicon dioxide) was added to the nanoemulsion and then dried in an oven (lines 113–114). The small-droplet oil phase should be adsorbed on silicon dioxide after water elimination. The dry powder of the final product was obtained from this process. Figure 1 shows the scanning electron microscopy (SEM) images of the dry powder emulsion as finished products.

The optical microscopy observation before drying process may not provide much information about product. Some of solubilized drug in oil phase may be converted to crystalline form during the drying process.    

Optical microscopy with polarized light is  the most easy-and-rapid to perform technique to  demonstrate the existence of crystals, although in the presence of amorphous solids . Other  techniques are obviously more complex and it is non-sense to apply them without this preliminar testing.

Response: Optical microscopy is not observable due to the presence of a large amounts of solid particles (silicone dioxide), which obscures transmitted light in of dry MIPEs. In particular, the small crystals of NP cannot be easily investigated by optical microscopy. Scanning electron microscopy, which has higher magnification and resolution, was chosen to observe the small-scale dry MIPEs. Moreover, DSC and PXRD analysis were used to confirm the presence of drug crystals.

Answer about the uncomplete dissolution do not solve the uncertainty: this fact is dued to a poor solubility or that a formulation  makes a cluster-effect retaining part of the drugdose.

Response: Thank you for your valuable advice. The discussion has been rectified in this revision. Line 331-333

Reviewer 3 Report (Previous Reviewer 2)

It's ok for publishing now.

Author Response

It's ok for publishing now.

Response: We really grateful for your reviewing the manuscript.

Round 2

Reviewer 1 Report (Previous Reviewer 1)

Most of the responses are satisfactory. 

Reviewer 2 Report (Previous Reviewer 3)

In agreement with the final version of the manuscript

This manuscript is a resubmission of an earlier submission. The following is a list of the peer review reports and author responses from that submission.

Round 1

Reviewer 1 Report

In the present manuscript entitled “Preparation and Characterization of Nifedipine-loaded Dry Medium Internal-phase Emulsions (Dry MIPEs) to Improve Cellular Permeability” authors have reported the dry emulsion to ultimately improve the oral bioavailability which has been mimicked by performing the cellular permeability in the present submission. I am not sure if the authors have chosen the right challenge to work on as nifedipine shows more than 90% bioavailability following the oral administration and is commercially available in the immediate release as well as extended release dosage form in tablet form for once in a day administration. I am not sure by designing the so-called Dry MIPEs which current problem can be overcome. Besides the poor rationale to justify the work there are many flaws/contradictions in the results which are as follows;

1.     The sizes presented in Table 2 are not matching with the sizes presented in the corresponding SEM images. There seems to be a huge difference in the size between these two.

2.     For most of the parameters in stability studies data is not shown which is very important to critically evaluate the author's findings.

3.     For PXRD analysis authors must have used the equivalent quantity of NP to have the comparison.

4.     What is the basis of performing the cellular uptake for 3 h when the dissolution studies were performed for 2 h? If the drug is getting dissolved in just half an hour (as per the author's findings) then for exact relation it could be better to perform these studies for 2 hours.

5.     Why the cellular uptake was performed only for 4 groups while all the groups were included in other studies. If authors finalized the 7RH8/32 as the optimized formulation then only this could be compared with NP.

6.     In the stability studies, authors have mentioned that precipitation of NP during storage may directly affect bioavailability. This may not be true in the present case as the authors have reported a dry emulsion in the present manuscript.

7.     As per the statement, in vivo drug absorption is affected by droplet diameter. The authors have not performed any in vivo study in the present submission. If we consider the cellular uptake as a possible substitute then also it is not conclusive as the used formulations for this study have almost the same particle size.

Reviewer 2 Report

Attached please see the suggestion!

Reviewer 3 Report

Please, read the atached Word-file

Reviewer 4 Report

The manuscript is very well written and very interesting. My question is - how do the resulting formulations relate to conventional nifedipine tablets?